# Planar Elongated B_12_ Structure in M_3_B_12_ Clusters (M = Cu-Au)

**DOI:** 10.3390/molecules28010236

**Published:** 2022-12-28

**Authors:** José Solar-Encinas, Alejandro Vásquez-Espinal, Luis Leyva-Parra, Osvaldo Yañez, Diego Inostroza, Maria Luisa Valenzuela, Walter Orellana, William Tiznado

**Affiliations:** 1Programa de Doctorado en Fisicoquímica Molecular, Facultad de Ciencias Exactas, Universidad Andrés Bello, Av. República 275, Santiago 8370146, Chile; 2Química y Farmacia, Facultad de Ciencias de la Salud, Universidad Arturo Prat, Casilla 121, Iquique 1100000, Chile; 3Facultad de Ingeniería y Negocios, Universidad de las Américas, Santiago 7500000, Chile; 4Grupo de Investigación en Energía y Procesos Sustentables, Instituto de Ciencias Químicas Aplicadas, Facultad de Ingeniería, Universidad Autónoma de Chile, Av. El Llano Subercaseaux 2801, Santiago 8900000, Chile; 5Departamento de Ciencias Físicas, Universidad Andrés Bello, Santiago 8370136, Chile; 6Computational and Theoretical Chemistry Group, Departamento de Ciencias Química, Facultad de Ciencias Exactas, Universidad Andrés Bello, Av. República 275, Santiago 8370146, Chile

**Keywords:** boron clusters, group 11 metals, potential energy surface, DFT computations, chemical bonding, aromaticity

## Abstract

Here, it is shown that the M_3_B_12_ (M = Cu-Au) clusters’ global minima consist of an elongated planar B_12_ fragment connected by an in-plane linear M_3_ fragment. This result is striking since this B_12_ planar structure is not favored in the bare cluster, nor when one or two metals are added. The minimum energy structures were revealed by screening the potential energy surface using genetic algorithms and density functional theory calculations. Chemical bonding analysis shows that the strong electrostatic interactions with the metal compensate for the high energy spent in the M_3_ and B_12_ fragment distortion. Furthermore, metals participate in the delocalized π-bonds, which infers an aromatic character to these species.

## 1. Introduction

Boron’s electronic deficiency allows it to assemble into diverse architectures to form systems with unusual electronic properties, making it promising for applications in diverse technologies. For instance, in super hard materials, semiconductors, nanomaterials, and compounds with potential biological applications [1,2,3,4,5,6,7,8,9,10].

However, this structural versatility makes it challenging to establish chemical bonding patterns that facilitate these systems’ structural rationalization. In contrast to its carbon neighbor in the periodic table, there are very limited models for predicting the structures of boron-based molecular systems. Boron hydrides, for instance, are characterized using electron counting-based rules [11,12,13,14,15,16,17,18].

Structures of bare B_n_ clusters have been thoroughly analyzed in recent years. These studies show that small clusters prefer planar structures up to a maximum size depending on the charge of the system. Cationic, neutral, and anionic clusters prefer planar structures up to 15, 20, and 40 atoms, respectively [19,20,21,22,23,24,25,26,27,28,29,30,31,32,33,34,35,36,37,38,39,40,41,42]. This demonstrates the significant effect that even one electron has on the structural preference of boron clusters.

In recent years, it has been shown that adding alkali metals to the B_12_ structure induces fascinating structural changes. For instance, in the LiB_12_ global minimum (GM) structure, the naked cluster B_12_ structure remains almost unchanged [43], but LiB_12_^–^ and NaB_12_^–^ GMs prefer a conical shape with an inner B_4_ ring and a higher concavity than the quasi-planar B_12_ cluster [44]. More noticeable is the D_6d_-Li_2_B_12_ GM, a tubular structure consisting of two stacked B_6_ rings capped with Li’s along the main axis [43]. Interestingly, the other alkali metals (Na-Cs) do not induce relevant B_12_ structural changes, highlighting the Li role [44]. An additional Li, in Li_3_B_12_, favors a cage-like B_12_ structure [43]. The isomerization energy decomposition analysis (IEDA) [45] has proven to be an appropriate theoretical method that provides an insightful explanation for these structural preferences; an enhanced stabilizing electrostatic interaction between the Li cations and the 3D-B_12_ moiety counterbalances the energetic cost of distorting the B_12_ fragment. A systematic review of metal-doped boron clusters is reported elsewhere [46].

What about the group 11 elements (Cu, Ag, Au)? Since they form compounds with oxidation states of +1, they could transfer one electron to B_12_, in analogy with the alkali metals. Our group has recently reported the minimum energy structures of CuB_12_ and CuB_12_^–^. In CuB_12_, Cu is placed capping the B_9_ ring of the B_12_-nacked cluster. In the anion, there are significant structural changes, with Cu participating in the electronic delocalization responsible for the doubly aromatic character of this species [47]. In AuB_12_ and AuB_12_, Au is bonded to one of the B’s of the B_12_ peripheral ring without inducing significant changes in the B_12_ structure [48]. All these results are consistent with the vast structural opportunities feasible by doping boron clusters with different metals [49,50,51,52,53,54,55,56], demanding systematic studies of the structural preferences for different combinations. This becomes a challenge from computational chemistry as the number and kind of atoms conforming to the study system increase since it demands the help of algorithms that facilitate the potential energy surface (PES) exploration [57,58].

Here, we show that doping B_12_ with three Cu, Ag, or Au atoms favors an elongated planar structure not preferred in the B_12_ bare cluster. Our study involves the PES exploration, employing genetic algorithms in conjunction with density functional theory (DFT) calculations, of the M_n_B_12_ combinations (M = Cu-Au, n = 1–3). Analysis of the chemical bonding and magnetic behavior shows that M_3_B_12_ are local and global aromatic species. The structural analogs to the Li_n_B_12_ global minima (n = 1–3) were also analyzed, showing that for n = 2 and 3, the structural preferences are drastically different. This highlights the importance of appropriate methods to provide reliable information on the energetically preferred structures of metal-doped boron clusters.

## 2. Results and Discussion

The putative GM structures of M_n_B_12_ (M = Cu-Au and n = 1–3) clusters are depicted in Figure 1, and relevant lowest energy isomers are depicted in Appendix A. Figure 1 shows that the B_12_ structure of the bare cluster is retained at the GM by adding one and two metal atoms. The first metal atom forms a bridged-like B-M-B bond with two boron atoms of the peripheral B_9_ ring (**1**-M), while the second one is placed on one of the edges of the MB_2_ triangle (**2**-M). Note that in previous reports the effect of doping B_12_ with other transition metals was evaluated, revealing that metal is placed on the main axis, above the peripheral B_9_ ring on the concave side of the B_12_ structure [49,50,51,52,53,54], which in the present work has been identified as an isomer close in energy to the putative global minimum of CuB_12_ (structure 1b, Appendix A). In the case of AgB_12_ and AuB_12_, this structure does not correspond to a local minimum since it has an imaginary frequency, and therefore is not reported. Furthermore, the analogous to the GM structures reported for the ZrB_12_ and AlB_12_ combinations [55,56] have relative energies higher than 20.0 kcal mol^–1^, thus lying above the relative energy range considered in our report. For the M_n_B_12_ (M = Cu-Au and n = 1–3) clusters, there are marginal changes in the B-B bond distances due to the addition of metals, more significant in the fragment where the metal is bonded. This agrees with the bond orders according to the WBI_B-B_ values, which are in the range of 0.4–1.3 in both the B_12_ and the M_n_B_12_ clusters (M = Cu-Au, n = 1, 2), where values close to or greater than 1.0 are among the peripheral atoms, as can be seen in Appendix A.

The WBI_M-B_ values are less than 1.0 for the bridged metal and close to 1.0 for the metal bonded to one boron at the periphery, while the WBI_M-M_ value in **2**-M is less than 0.1, showing that metals lack a significant covalent bonding character. When one more metal is added, in M_3_B_12_, the minimum energy structure has a significantly different B_12_ fragment than that of the B_12_ GM. An elongated planar structure is now favored, **3**-M (Figure 1). The stabilizing role of M to tend planar elongated structure is significant, as evidenced in Figure 2, which shows how the relative energy of this motif decreases as M atoms are added to become the GM in M_3_B_12_. WBI_B-B_ values range from 0.4 to 1.4, with values near or above 1.0 among the outer borons. One of the M atoms closes the elongated ring, while the other two are linked at the periphery in bridged positions with the M-B edges, the three metal atoms forming a linear M_3_ structure. The WBI_M-B_ values between the extreme M and B are the highest, more significant than 0.5, while the WBI_M-B_ values involving the central M are lower (see Appendix A). The charges from the natural population analysis (NPA) indicate a charge transfer from the M to the B_12_ fragment (q_M_ = 0.2–0.7 |e|); however, it does not reach the magnitude of the alkali atoms, where almost one electron per metal atom is transferred (Appendix A). Additionally, the natural electron configurations of the Cu, Ag, and Au atoms in these systems are reported in Appendix A. The charge transfer occurs mainly from the 4s and 5s orbitals of Cu and Ag, respectively. While for systems with Au, this transfer, to a lesser extent, occurs from the 5d orbitals.

After reviewing the identified structures, we realized that the counterpart structures of the Li_n_B_12_ (n = 2, 3) GMs were missing. This prompted us to evaluate these geometries, and as shown in Figure 3, the relative energies when Li is replaced by group 11 metals change significantly. This underlines the importance of using suitable algorithms to explore the PES since it is unreliable to use structural knowledge of related systems, which could lead to entirely wrong conclusions. The structural counterparts are local minima, except for Au_2_B_12_ and Au_3_B_12_, with two and one imaginary frequencies. Remarkably, for M = Ag and Au, the relative energies are at least 50.0 kcal mol^–1^ above the GM identified by PES exploration.

To gain further insights into the chemical bonding, we performed an AdNDP analysis. Orbital localization methods proved insightful in understanding chemical bonding in boron clusters [59,60]. The results of the AdNDP analysis for all systems are reported in Appendix A. In the three systems **1**-M, **2**-M, and **3**-M, five, ten, and fifteen 1c-2e lone pairs are identified, corresponding to the d orbitals of the metals of each system. For **1**-M and **2**-M, the bonding picture of the B_12_ fragment remains identical to that of the bare B_12_ cluster (see Appendix A). In **1**-M, the metal connects to the B_12_ fragment through a three-center one-electron (3c-1e) B-M-B σ-bond, as shown in Appendix A.

In the case of **2**-M, the bonding situation is very similar, only now the second M atom is attached to one of the periphery B’s by one M-B 2c-2e σ-bond. Furthermore, the bridged M is now linked via a B-M-B 3c-2e σ-bond at the cost of delocalization of the periphery B-B bond. The **3**-M bonding will be discussed in more detail. For **3**-Cu, the localized orbitals, excluding the lone pairs, recovered by AdNDP are reported in Figure 4. The other systems show similar bonding patterns (see the full bonding picture in Appendix A). Eight B-B 2c-2e σ-bonds link the nine periphery borons. Seven 3c-2e σ-bonds are also detected between each triangular fragment that forms the three internal boron with those of the periphery. The M extremes of the M_3_ fragment are linked to borons by M-B 2c-2e σ-bonds. The central M does not participate in any bonds, according to AdNDP. The remaining electrons, AdNDP locates in three π-orbitals, two of 6c-2e, and one of 4c-1e, for a total of five π-electrons. These results suggest the possibility of (anti)aromaticity, local or global, given both σ- and π-delocalized bonds. We will discuss below this possibility by analyzing the **3**-M magnetic behavior.

The shape of the **3**-M structure evokes the geometry of an isomer of its valence isoelectronic B_13_ cluster, which is 3.7 kcal mol^–1^ above the GM (at the PW91/TZ2P level) [61]. The AdNDP analysis of this system is shown in Figure 5, where it is seen that there are significant differences with **3**-M, the B_9_ contour is closed by 10 B-B 2c-2e σ-bonds, there are six delocalized 4c-2e σ-bonds connecting the internal B_3_ fragment to the B_9_ contour, and AdNDP places seven electrons forming π-bonds. These bonding differences, i.e., having 4n delocalized σ-bonds, could be responsible for the non-planarity of B_13_.

The isomerization energy decomposition analysis (IEDA) allows us to analyze quantitatively the structural preference between two isomers in terms of energy components [45]. A hypothetical thermodynamic cycle used to perform the IEDA in the M_3_B_12_ systems is shown in Appendix A. Thus, we compare **3**-Cu vs. the first Cu_3_B_12_ isomer that preserves the B_12_ fragment of the naked cluster (ΔE_iso_ = 13.0 kcal mol^−1^); the relative energy is slightly different from that reported in Appendix A since the IEDA calculations were performed at the PBE0-D3-BJ/ZORA/TZ2P level. The values summarized in Table 1 show that interactions favor GM in the orbital (ΔΔE_orb_ = 115.3 kcal mol^–1^) contribution, compensating for the electrostatic (ΔΔV_elstat_ = –7.9 kcal mol^–1^) and Pauli repulsion (ΔΔE_Pauli_ = –19.5 kcal mol^–1^) terms which favor the higher energy isomer; the difference in the dispersion interaction is negligible. These results show that the covalent interactions between M_3_ and the B_12_ are enhanced and are more important in terms of magnitude than ionic interactions in the GM, thus accounting for its preference. However, ΔE_iso_ also depends on the distortion energy of each fragment, favoring, in this case, the higher energy isomer (ΔΔE_dist (total)_ = –74.5 kcal mol^–1^). Note that the major contribution of this term comes from the distortion of the M_3_ fragment, while the B_12_ fragment slightly prefers the geometry it has in the GM. Therefore, Cu_3_B_12_ prefers the planar elongated shape granted by better orbital interactions between the Cu_3_ and B_12_ fragment, compensating for the energy cost to distort its M_3_ moiety, and to a lesser extent, the unfavorable electrostatic interaction and Pauli repulsion terms. IEDA predicts a similar trend for Ag_3_B_12_ with the difference that the Pauli repulsion term, in this case, also favors GM. Finally, for Au_3_B_12_, the results are slightly different. The distortion energies follow the same trends and magnitudes as for its lighter analogs, and the orbital interaction remains the most important term; however, the electrostatic interaction, which, as in Cu_3_B_12_ and Ag_3_B_12_, favors the second isomer, is now comparable in magnitude to the orbital term, so that the balance of all the terms that make up ΔE_iso_ leaves both isomers virtually isoenergetic at the PBE0-D3-BJ/ZORA/TZ2P level. Note that although at the PBE0-D3/def2-TZVP level, the planar elongated system is still predicted to be the global minimum of Au_3_B_12_, the difference in energy between it and the highest energy isomer is much smaller (2.1 kcal mol^–1^) than for Cu_3_B_12_ and Ag_3_B_12_.

AdNDP analysis reveals the presence of delocalized π-bonds, which could be associated with a possible (anti)aromatic character. To investigate this possibility, we have evaluated the current densities induced by an external magnetic field (perpendicular to the molecular plane). According to the magnetic criteria, an (anti)aromatic system is characterized by the presence of (diatropic) paratropic ring currents circuits. Figure 6 shows that **3**-Cu exhibits both local and global diatropic ring currents circuits. The local circuits are internal and around the inner B atoms. At the same time, the global ring current surrounds the cluster decorating the external M_3_B_9_ chain. By analyzing the integrated current profiles (Appendix A), it has been possible to estimate the ring current strength (RCS) of each circuit (see Figure 6). RCS values are significant, considering benzene has a net RCS of 11.8 nA T^–1^ at the same level. The magnetic behavior of **3**-Ag and **3**-Au are similar, as shown in Appendix A. Therefore, the **3**-M are classified as local and global aromatic species according to the magnetic criterion.

## 3. Materials and Methods

We systematically explored the potential energy surfaces employing the AUTOMATON program [57,62], with an initial screening (in the singlet states) at the PBE0 [63]/SDDALL [64] level. The low-lying energy isomers (<20.0 kcal mol^–1^ above the putative global minimum) were re-minimized at the PBE0-D3 [65]/def2-TZVP [66] level. The top isomers were optimized in the triplet state at the PBE0-D3/def2-TZVP level to test the relative energies at this multiplicity. To provide more reliability in our energetic analysis, relative energies were computed using the domain-based local pair-natural orbital-based single-, double-, and perturbative triple excitations coupled cluster DLPNO-CCSD(T) method [67] as implemented in ORCA-4.2.1 [68,69] in conjunction with extrapolation to the complete basis set limit via the def2-SVP and def2-TZVP basis sets [70,71] (labeled as DLPNO-CCSD(T)/CBS). This refinement was performed for isomers up to 10.0 kcal mol^–1^ above the putative global minimum. The chemical bonding was analyzed (at the PBE0-D3/def2-TZVP level) using the Wiberg bond index (WBI) and natural population analysis (NPA), as implemented in the NBO 6.0 program [72]. Furthermore, the adaptive natural density partitioning (AdNDP) method [73,74] was performed with the Multiwfn program [75]. AdNDP represents the electronic structure in n-center-two-electron (nc-2e) bonds, with n ranging from one to the total number of atoms in the molecule, recovering the Lewis’ electron pair concept as the fundamental chemical bonding component complemented with delocalized bonds, when they are present. Isomerization energy decomposition analysis (IEDA) [45] was computed. IEDA allows the decomposition of the isomerization energy (ΔE_iso_) in terms of the distortion energy of the fragments (ΔE_dist_) and the change in the interaction energies between the fragments of each isomer (ΔΔE_int_). The latter term, in turn, is decomposed as the sum of the changes in the orbital (ΔΔE_orb_) and electrostatic (ΔΔV_elstat_) interaction, the Pauli repulsion (ΔΔE_Pauli_), and the dispersion energy (ΔΔE_disp_). This analysis was performed at the PBE0-D3-BJ [76]/ZORA [77]/TZ2P [78] level using ADF2012 [79].

To assess aromaticity, we have analyzed the current densities induced by an external magnetic field applied perpendicularly to the molecular plane (at the PBE0-D3/def2-TZVP level) using the GIMIC program [80,81], which employs the gauge-included atomic orbitals (GIAO) method [82]. For vector plots, we used the Paraview 5.10.0 software [83,84]. The ring current strengths (RCS), a quantitative descriptor of aromaticity, were obtained by integrating the ring current flow in a perpendicular plane using the two-dimensional Gauss–Lobatto algorithm [80,85] as implemented in GIMIC. The integration planes correspond to cut-off planes perpendicular to the chosen bonds of the interest annular moiety and extend horizontally for 3.6 Å along the ring’s plane, with 2.6 Å above and below the bond. Positive (diatropic), negative (paratropic), and near-zero RCS values indicate aromaticity, antiaromaticity, and non-aromaticity. The bisected current densities (sigma and pi) were calculated using AIMAll software [86] at the PBE0-D3/def2-TZVP level. As previously established in other studies, the different ring current circuits have been identified by analyzing the RCS profiles in appropriate planes [87,88].

## 4. Conclusions

The effect of doping the B_12_ cluster with group 11 metals, in M_3_B_12_ (M = Cu-Au) clusters is evaluated to compare with the impact of alkali metals, especially Li, where the minimum energy structures of Li_2_B_12_ and Li_3_B_12_ consist of B_12_ arrays quite different from those of the bare B_12_ cluster. Exploration of the potential energy surface, using genetic algorithms and DFT calculations, reveals that adding one or two metal atoms induces substantial changes in the B_12_ fragment, with minor modifications in bond distances, especially of the fragments in contact with the metals. However, when the third M atom is added, in M_3_B_12_, a structure containing a planar elongated B_12_ fragment (**3**-M), whose shape is high-energy in the bare cluster, is favored. This is evidence of the wealth of structural possibilities by doping boron clusters with metals, where the nature of the metal may have unexpected effects. The latter also evidences the need for adequate methods to identify minimum energy structures since no structural preferences can be assumed based on similar systems. For example, the GM analogous structures of Li_2_B_12_ are less stable by at least 7 kcal mol^–1^, while the Li_3_B_12_ analogs are not even all local minima and are more than 50 kcal mol^–1^ away from the most stable one. Bonding analysis evidences the presence of delocalized σ- and π-bonds in **3**-M. Furthermore, magnetically induced current density analysis evidences the existence of local and global diatropic ring currents, characterizing them as aromatic.

## Figures and Tables

**Figure 1 molecules-28-00236-f001:**
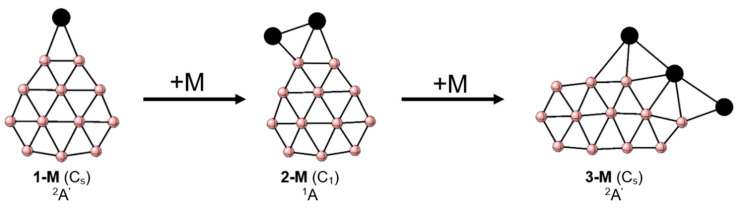
Lowest energy structures for M_n_B_12_ clusters (M = Cu-Au and n = 1–3) at the PBE0-D3/def2-TZVP level. M: black spheres, B: pink spheres.

**Figure 2 molecules-28-00236-f002:**
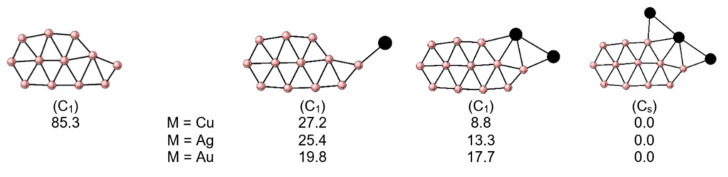
Stabilization of the elongated planar B_12_ fragment as it is doped with M atoms in M_n_B_12_ clusters (M = Cu-Au and n = 1–3). Structures were optimized at the PBE0-D3/def2-TZVP level. The relative energies are also reported in kcal mol^–1^ (regarding GM for each combination). M: black spheres, B: pink spheres.

**Figure 3 molecules-28-00236-f003:**
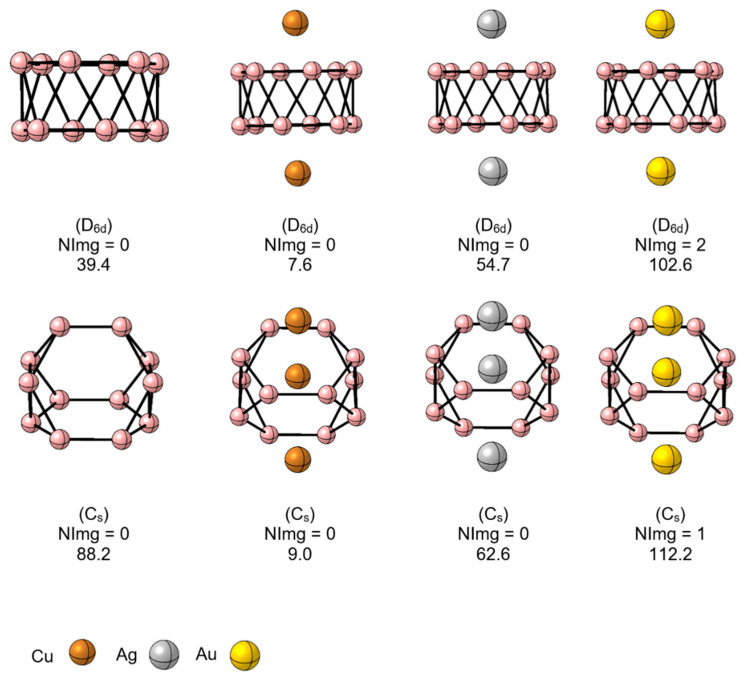
Optimized structures for M_n_B_12_ (M = Cu-Au and n = 2, 3) in analogous geometries to the global minima of Li_2_B_12_ and Li_3_B_12_ clusters, and their relative energies (kcal mol^–1^) concerning the corresponding GMs, at the PBE0-D3/def2-TZVP level.

**Figure 4 molecules-28-00236-f004:**
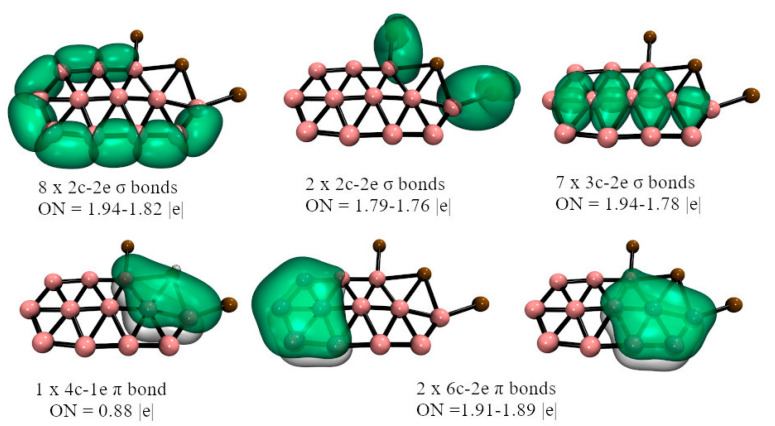
Chemical bonding representation according to the AdNDP method of **3**-Cu, at the PBE0-D3/Def2-TZVP level. Cu: brown spheres, B: pink spheres.

**Figure 5 molecules-28-00236-f005:**
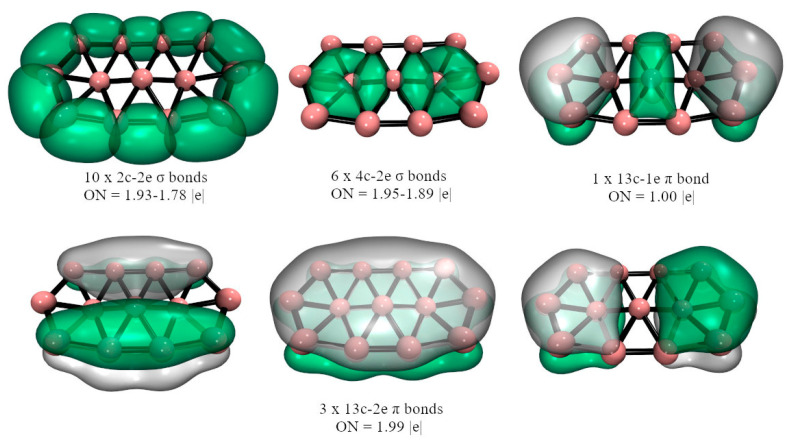
Chemical bonding representation according to the AdNDP method of B_13_, at the PBE0-D3/Def2-TZVP level.

**Figure 6 molecules-28-00236-f006:**
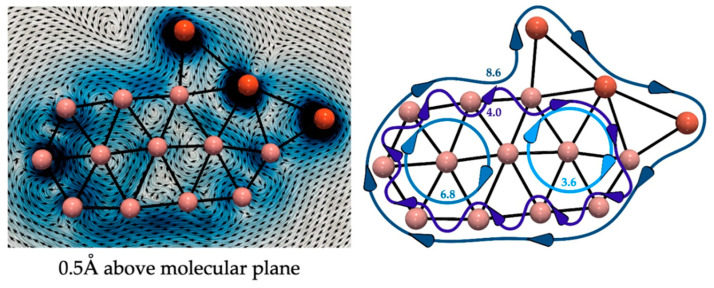
On the left, the magnetically induced current density of Cu_3_B_12_ in a plane placed 0.5 A above the molecular plane. Diatropic currents are assumed to circle clockwise, and paratropic ones circle anticlockwise. On the right are the detected currents’ paths and their strength (in nA T^–1^).

**Table 1 molecules-28-00236-t001:** IEDA results (kcal mol^–1^) at the PBE0-D3-BJ/ZORA/TZ2P level for the M_3_B_12_ clusters with M_3_^3+^ + B_12_^3–^ as fragments.

System	Cu_3_B_12_	Ag_3_B_12_	Au_3_B_12_
ΔE_iso_	13.0	9.8	−0.2
ΔE_dist_ (M_3_^3+^)	−82.1	−75.7	−73.2
ΔE_dist_ (B_12_^3−^)	7.6	6.7	8.2
ΔΔE_int_	87.5	78.9	64.8
ΔΔE_orb_	115.3	85.2	86.8
ΔΔV_elstat_	−7.9	−22.5	−70.6
ΔΔE_Pauli_	−19.9	16.4	49.0
ΔΔE_disp_	0.0	−0.2	−0.4

## Data Availability

Not applicable.

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
