# Peer review of "Planar Elongated B12 Structure in M3B12 Clusters (M = Cu-Au)"

_molecules, 2022, doi:10.3390/molecules28010236_

Round 1

Reviewer 1 Report

The paper describes a procedure for finding the most stable strcuture of M3B12 clusters. The authors used a global search algorithm in order to find the lowest total energy states and found them possess unusual elongated planar structures. The work is interesting in terms of most recent research interest in Boron-based low dimension materials.

My concerns include the following points:

(1) In the section 'results and discussion', one expects a serious comparison between the present results and the bonding in CuB12, CuB12, AuB12 and AuB12 of the literature listed the manuscript.

(2)There are a few published articles about structural and electronic properties of metal atom doped B12 cluster that could be included in the introduction of this manuscript: RSC Adv., 2016, 6, 27177-27182; RSC Adv., 2019, 9, 2870–2876; ACS Omega 2020, 5, 20525−20534; J. Phys. Chem. A 2014, 118, 80988105; Phys. Chem. Chem. Phys., 2018, 20, 23740—23746; J. Phys. Chem. C 2019, 123, 6276−6283; J. Phys.: Condens. Matter 2022. 34, 445302; Inorg. Chem. 2018, 57, 343−350. Author should perform comparison the structures between these clusters and the present results.

(3) The energy difference among the low-lying energy isomers is small. To further refined the ground state structure, the single-point energies calculated using the more accurate CCSD(T) method should be conducted.

(4) Cu, Ag and Au are transition metal atoms. Lone pairs of d orbitals can be expected in AdNDP analysis, especially for these peripheral metal atoms. But it can not be seen. The author should explain it.

(5) To further reveal chemical bonding properties, especially for M-B bond, electron localization function (ELF) or localized orbital locator (LOL) is suggested to given.

(6) More detail of charge transfer is encouraged to given, especially for s, p and d orbitals of metal atoms. We can know the electron configuration and oxidation state of metal atoms from these results.

Author Response

We are attaching our response.

Reviewer 2 Report

The authors studied the M3B12 (M=Cu, Ag, and Au) clusters from the view of geometric and electronic structures using genetic algorithms and density functional theory calculations. They give the conclusion that M3B12 have planar elongated B12 structure and show aromatic character. On the whole, the employed computation levels and the discussion in this work are moderate. In my opinion, there are still some problems need to be solved if this paper can be accepted.

1.      It is confused that structures of M1B12 and M2B12 in Figure 1 are different from that of M1B12 and M2B12 in Figure 2. Another problem, the relative energies in Figure 2 are the difference of electronic energies for each molecule? If yes, those values are no meaning because the M3B12 must have the lowest energy due to their most electrons.

2.      For those analogous geometries of Li2B12 and Li3B12, which geometry is zero point for the relative energies? This should be pointed out. As above question, what is definition of relative energy?

3.      The authors optimized all the possible structures including cluster and planar for M2B12 and M3B12, as they presented these clusters in Figure 3. However, in the supporting information, there are no information of structures and energies for Ag2B12, Au2B12, Ag3B12, and Au3B12 cluster in Figure 5, 6, 8, and 9, so how to compare the stability between cluster and planar?

4.      For the AdNDP analysis, although the calculated results are no problem, the discussion about chemical bonding are in doubt. As depicted in Figure 5, the first and third 13c-2e are not π bonds. Form the orbitals character, they are anti-π bonds.

5.      There are still careless places, such as the caption of Figure 4, “1-Cu” should be “3-Cu”? The authors should check the writing thoroughly.

Author Response

We are attaching our response.

Round 2

Reviewer 1 Report

no

Reviewer 2 Report

The authors replied all the questions from reviewer. The paper can be published in current form.